# Intuitive assessment of spatial navigation beyond episodic memory: Feasibility and proof of concept in middle-aged and elderly individuals

Sophia Rekers [1,2,3]*, Michael Niedeggen[2]

1 Berlin School of Mind and Brain, Humboldt-Universität zu Berlin, Berlin, Germany, 2 Department of Education and Psychology, Freie Universität Berlin, Berlin, Germany, 3 Department of Neurology with Experimental Neurology, Charité–Universitätsmedizin Berlin, Corporate Member of Freie Universität Berlin, Humboldt-Universität zu Berlin, Berlin, Germany

* sophia.rekers@hu-berlin.de

**Data Availability Statement:** The navigation assessment and the script used to create the here presented analyses are available on the Open Science Framework at https://osf.io/z5waq/ and the

## Abstract

Deficits in spatial navigation in three-dimensional space are prevalent in various neurological disorders and are a sensitive cognitive marker for prodromal Alzheimer's disease, but are also associated with non-pathological aging. However, standard neuropsychological tests used in clinical settings lack ecological validity to adequately assess spatial navigation. Experimental paradigms, on the other hand, are often too difficult for seniors or patients with cognitive or motor impairments since most require operating a human interface device (HID) or use complex episodic memory tasks. Here, we introduce an intuitive navigation assessment, which is conceptualized using cognitive models of spatial navigation and designed to account for the limited technical experience and diverging impairments of elderly participants and neurological patients. The brief computer paradigm uses videos of hallways filmed with eye tracking glasses, without employing an episodic memory task or requiring participants to operate a HID. Proof of concept data from 34 healthy, middle-aged and elderly participants (56–78 years) provide evidence for the assessment's feasibility and construct validity as a navigation paradigm. Test performance showed normal distribution and was sensitive to age and education, which needs to be considered when investigating the assessment's psychometric properties in larger samples and clinical populations. Correlations of the navigation assessment with other neuropsychological tests confirmed its dependence on visuospatial skills rather than visual episodic memory, with age driving the association with working memory. The novel paradigm is suitable for a differentiated investigation of spatial navigation in elderly individuals and promising for experimental research in clinical settings.

data used in this study is available on the open access repository of the Humboldt-Universität zu Berlin at https://doi.org/10.18452/22757.

**Funding:** The authors received no specific funding for this work.

**Competing interests:** The authors have declared that no competing interests exist.

# Introduction

The ability to find our way from one place to another and to position ourselves in a three-dimensional environment, also called topographical or spatial navigation, is of great significance for people's autonomy and safety. Spatial disorientation and navigation deficits are prominent in various neurological disorders and have been recognized as an early indicator of Alzheimer's disease (AD) [1–3] and multiple sclerosis [4]. Furthermore, orientation and navigation deficits have been used to differentiate between different types of dementia, for instance AD and frontotemporal dementia [5, 6].

Independent of specific diseases, it has been repeatedly shown that older age itself is associated with a decrease in navigation performance [7–9]. There is evidence, that the age effect in navigation performance also goes beyond deficits in memory systems and is in part driven by difficulties in forming and using cognitive representations [10] and adapting navigational strategies [11]. However, the exact nature of the interplay of different computations involved in age related decline of spatial navigation remains elusive [12]. From a neurobiological standpoint, spatial navigation has previously been related to activation in the medial temporal lobe and parietal cortex [13], and both are among the first brain regions affected by pathological changes due to AD [14, 15]. Using immersive virtual reality (VR), Stangl et al. [16] provided evidence of impaired path integration and less stable representations of grid cells, specialized cells in the medial temporal lobe for the spatial coding of environments, in healthy older adults compared to younger adults. Furthermore, the magnitude of grid cell representations, but not standard neuropsychological tests, were associated with path integration performance in older adults.

Despite the evidence indicating the usefulness of assessing spatial navigation in older adults, no gold standard has been defined in the examination of spatial navigation, either experimentally or clinically. The heterogeneity of available navigation paradigms does not only stem from its functional complexity, but also from the fact that spatial navigation requires large-scale three-dimensional environments. VR paradigms with different degrees of immersion allow for a wider application of navigation tasks in different settings and can provide detailed information on spatial navigation abilities in a more standardized and ecologically valid way [17]. However, active navigation in a virtual environment without body-based cues such as vestibular and proprioceptive information can be challenging for participants with little experience using human interface devices (HID) to move (e.g. keyboard, joystick). Consequently, gaming experience significantly influences the performance in a navigation task [18]. Another constraint is related to the mismatch between moving visual input and stationary motor input during locomotion in a virtual environment, which can increase the chance of adverse effects such as motion sickness (i.e. cybersickness) and impact the feasibility of a VR paradigm [19].

Considering the restrictions mentioned above, clinical-neuropsychological examinations of spatial navigation are mostly based on tests reflecting subcomponents of this cognitive function. Common examples are a patients' knowledge about their current location (e.g. spatial orientation in the Mini–Mental State Examination; MMSE) [20]; visuoconstructive tasks such as the Rey-Osterrieth Complex Figure Test (ROCF) [21] and the Clock-drawing test [22]; mental rotation tasks such as Vandenberg's mental rotation test [23]; and "left-vs-right" decision tasks such as Money's Standardized Road-Map Test [24]. Other approaches to approximate spatial navigation use visual episodic memory tests that require memorizing and recalling routes on a map (e.g. Learning and Memory Test [Lern- und Gedächtnistest; LGT 3) [25]. In more naturalistic approaches, such as the route learning item used in the Rivermead Behavioural Memory Test (RBMT) [26], participants are asked to memorize and recall a sequence of movements in the examination room, which differs in size between test sites and thus limits

generalizability of the result. While these neuropsychological tests have low demands on the technical equipment and patients' abilities, they fail to incorporate essential aspects of spatial navigation, such as involving spatial updating and both egocentric (observer-based) and allocentric (survey) perspectives.

For the development of the novel paradigm, we envisaged assessments of elderly participants in clinical and research settings as the area of application. Thus, the practical considerations for the navigation assessment included a reasonable application time that would allow the test to be included in a time-restricted neuropsychological examination as well as short trials and individually adjusted, frequent pauses that would decrease the likelihood of motion- or cybersickness related symptoms, which have been associated in VR paradigms with increased exposure time [27, 28]. Furthermore, trials should gradually increase in complexity to prevent dropout due to excessive or insufficient cognitive demand and the assessment's application and evaluation should be standardized, to allow for maximum comparability between administrators and test locations.

A high priority was the ecological relevance and intuitiveness of the task. To strengthen the latter, we refrained from an active navigation/ exploration task, which incorporates the possibility to choose one's own path, and opted for a passive navigation paradigm to ensure that operating a HID does not interfere with assessment duration or performance. Furthermore, passive navigation has been shown to be easier for older participants [29], and observing another person navigate has been shown to induce similar activation patterns in the medial temporal lobe than navigating oneself [30]. As an ecologically relevant setting, we chose the task of finding specific doors in hallway environments. Lastly, and most importantly, the paradigm needed to be valid and represent spatial navigation in compliance with cognitive models of spatial navigation.

To increase construct validity, we conceptualized the paradigm closely along the lines of the well-established model of factors that impact spatial navigation ability by Wolbers and Hegarty [31]. Following this model, we made deliberate choices concerning which spatial cues, computational mechanisms, and spatial representations should be integrated in an intuitive navigation assessment:

*Spatial cues* used to navigate should be naturalistic and relevant. Thus, we decided to include geometric structures and local landmarks with high ecological relevance, while providing symbolic representations that are intuitive to understand, even for inexperienced navigators. To ensure few requirements on technical and time capacities at the test site and low demands on the motor abilities of the participants, we restricted self-motion information to optic flow. While non-visual input such as vestibular cues, motor efference copies and proprioceptive feedback are important cues in everyday life to estimate one's position and orientation [32], the implementation of multimodal input in test settings would require physical movement by the participant, such as the approach used by Stangl et al. [16] and/or advanced technical setups, for example, using treadmills [33]. Notably, treadmill approaches can increase the number of dropouts compared to desktop assessments [34], and visual information from optic flow has been shown to be a sufficient self-motion cue for accurate spatial updating [35].

*Computational mechanisms* involved in a navigation task can be categorized according to four interrelated cognitive domains, which we aimed to integrate in the novel task: (1) visuo-perceptive/constructive functions; (2) short-term and working memory; (3) mental rotation and perspective taking; and (4) executive functions. These four functions affect the computational processes involved in navigation in different ways. For instance, working memory, and to a lesser extent short-term memory, play a critical role in using spatial cues to update one's position by extrapolating from egocentric and/or allocentric movement to estimate the

distance and direction travelled [36]. Both spatial updating and working memory have been linked to frontoparietal and especially precuneus activation [37, 38]. Moreover, mental rotation ability becomes essential when shifts in perspective center around objects while perspective taking requires egocentric transformations [39]. We decided to operationalize shifting spatial perspectives by requiring updating under rotated perspectives.

We limited the integration of executive processes beyond working memory to those that can be meaningfully integrated into a passive navigation task and to keep the strategic approach to solving the task homogenous across participants. Thus, we decided against implementing goal selection or route planning, since these require that individuals make decisions about the route they take. However, functions that prevent perseveration such as novelty detection, uncertainty resolution and resetting of previous information are essential for successful navigation [40]. To implement this in the novel assessment, we used homogenous yet not identical objects and environments.

*Spatial representations* used in a navigation task may be represented egocentrically or allocentrically and can be available throughout the task (online) and/or retrieved, recognized or inferred from previously encountered representations (offline). Considering that older age has been associated with difficulties in forming and using cognitive representations [10], we decided to include a symbolic representation of the environment's allocentric perspective instead of requiring the formation of cognitive maps by the participants. This was done to make the task easier but, more importantly, to decrease the episodic memory component of the task.

On the basis of these theoretical considerations, a parsimonious novel paradigm was developed, which should allow a standardized assessment of spatial navigation. In contrast to other assessment procedures, the novel paradigm relies less on individual preferences and experiences for strategies, especially the use of local versus distal cues and a sequential egocentric versus allocentric, cognitive map approach. By presenting egocentric and allocentric reference frames simultaneously it focuses on the process of online spatial updating in successively more complex environments, which in turn reduces the load on episodic memory.

Beyond theoretical conceptualization, empirical evidence is necessary to test the construct validity of the novel navigation assessment. Although indicators of concurrent validity may be limited for novel test concepts, indicators of convergent validity must be tested using neuropsychological tests of the subcomponents of the test concept described. For the novel navigation assessment convergent validity indicators are correlations with markers for visuospatial tests, short-term and working memory, mental rotation and perspective taking. Discriminant validity is indicated by looking at markers for episodic memory and executive tests on planning and strategy application. An empirical assessment of neuropsychological tests whose performance is associated with or distinct from the test performance in the navigation assessment could help to identify factors that require modification.

In this work, we investigated the feasibility of the novel paradigm in healthy, elderly participants. Along this line, we examined whether any non-predefined factors led to dropouts or outliers. Furthermore, we investigated test score distribution and relationship with demographic factors, specifically age, that need be considered in future assessments of psychometric properties in larger samples. Lastly, we aimed to learn more about the relationship of the task performance with a selection of cognitive variables and its dependence on demographic factors to test the implementation of the theoretical concept and provide empirical evidence for the construct validity of the navigation assessment. Specifically, we intended to investigate whether test performance was associated with visuospatial rather than episodic memory markers and how this association related to participant age.

## Methods

### Participants

44 participants were recruited via mailing lists of the local university's auditors' program and a local senior's university program. Only participants with normal or corrected-to-normal spatial vision (two included participants reported having impaired color vision) and no history of neurological or psychiatric disorders were included. The short version of the Geriatric Depression Scale (GDS) [41] was applied to screen for participants scoring above 11 out of 15 points, indicating a severe depression [42]. Furthermore, to account for potential self-selection bias, we only included participants without meaningful subjective cognitive complaints. This was operationalized by an average score below 2, indicating, that problems occur less frequently than "sometimes," in the Complainer Profile Identification (CPI) [43] or a subjective spatial orientation questionnaire adapted from the self-report measure of spatial orientation strategies (Fragebogen zu Räumlichen Strategien; FRS) [44]. Details on the applied measures can be found in S1 File.

To screen for general cognitive impairment, we administered the MMSE. All participants scored 28 points or more, which is above the threshold indicating cognitive impairment. To screen for visuospatial impairments, the screening task "Line division," which assesses length perception, from the module "Space" of the Computer-based Assessment of Visual Functions (CAV) [45] and the navigation assessment's pre-test on perspective translation, described below, were applied. The personal orientation test by Semmes, Weinstein, Ghent, and Teuber [46] was applied to screen for basic orientation on one's own body and left-right orientation. 10 participants were excluded for the following reasons: neurological history (strokes) (n = 2); meaningful subjective cognitive complaints (n = 6); one participant did not pass the perspective translation pre-test test likely due to severe strabismus; and one participant did not pass the personal orientation test.

The sample for statistical analysis consisted of 34 right-handed individuals (25 female), whose average age was of 68.50 years (*SD* = 5.77, range 56–78 years). All but one participant spoke German as their native language and approximately half (52.94%) lived with their partner or relative, while the other half lived alone or in another independent living arrangement. The majority of participants had a high school qualification to attend university (61.76%); 29.41% held a secondary school certificate and 8.82% held an upper secondary school certificate. Furthermore, the majority reported to be pensioners (82.35%). Participants received either financial compensation of 7€ or alternatively a report on their results in the cognitive tests. The study design was approved by the ethics committee of the Department of Education and Psychology at the Freie Universität Berlin.

### Cognitive tests for the assessment of construct validity

The cognitive assessment comprised seven tests, including the novel navigation task and screening tests. Spatial updating and manipulation of short-term spatial information in online representations, were operationalized with the visuospatial short-term and working memory tasks visual memory span forward and backward from the Wechsler Memory Scale-Revised Version (WMS-R) [47]. Retrieval and recognition of visual information were included to represent use of offline representations for navigation and operationalized with the visual episodic memory tasks "Visual Learning" for immediate recall and "Delayed Recognition: Figures" from the Inventory for Memory Diagnostics (Inventar zur Gedächtnisdiagnostik [IGD]) [48]. Finally, mental rotation ability was assessed by the subtest "Rotation" from the module "Space" of the CAV to operationalize the spatial computation of shifting perspectives.

## Procedure

The participants were tested individually in a testing room of the local neuropsychological ambulance in one 45-to-60-minute session, which included consent forms, questionnaires, neuropsychological assessment and the novel navigation assessment. After giving written and informed consent, the participants were assigned a pseudonymized code, and asked for their demographic information and their medical history regarding factors relevant for study inclusion. Subsequently, questionnaire measures, which were sent to the participants before the examinations, were verified for completeness and open questions were clarified. Following, a questionnaire concerning subjective complaints in spatial orientation abilities was administered. Afterwards, the cognitive tests were administered in the following order: WMS-R block-span tests forward and backward; novel navigation assessment; IGD subtest on visual learning; MMSE (without figure drawing); Semmes' personal orientation test; IGD subtest visual recognition; MMSE figure; CAV subtests "Line division" and "Rotation."

**The novel navigation assessment.** To emulate navigating in an unfamiliar environment, each trial of the navigation assessment consists of a video of an egocentric exploration of a different hallway and a schematic map of the explored hallway (Fig 1). The map indicates the starting point, the structure of the hallway, the position of the doors and the position of stairs without further landmarks. The videos were recorded from the point of view of a person walking along the hallway. In the beginning of each video, the person shortly examines the corridor by looking left and right from the starting point before starting to walk with a pace of approximately two steps per second, while looking ahead. At each door and each intersection, the person in the video looks directly at the handle of the door or down the hallway before continuing straight ahead or turning into a corridor. The task for the participant is to mentally trace the position of the person in the video on the map, and to decide which of the doors on the allocentric map corresponds to the door chosen in the video. The answer is documented by the examiner on an answer sheet.

Updating in the form of short-term and working memory, and the synchronization of both the egocentric and allocentric perspective are implemented in the task with the cognitive tracing of the person's position in the video on the map. To include perspective rotation, trials were incorporated in which the person in the video takes 90° turns at intersections or turns 180°, while the allocentric map remains in the original position. In those trials, participants must continue tracing the position of the person in the video in a rotated perspective. No verbal cues on the extent or direction of rotations were included with the purpose that participants must use the visual input to trace the position of the person correctly. To minimize the

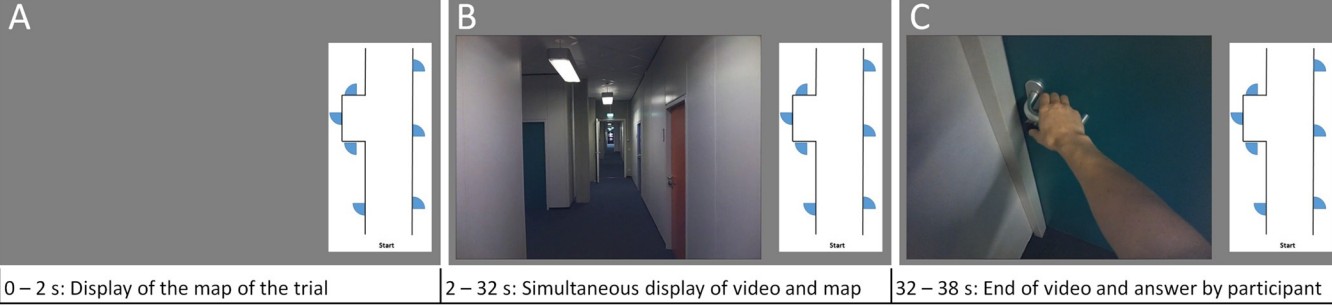

| A | B | C |
|---|---|---|
| 0 – 2 s: Display of the map of the trial | 2 – 32 s: Simultaneous display of video and map | 32 – 38 s: End of video and answer by participant |

**Fig 1. Schematic representation of the instruction trial.** (A) Each trial starts with the display of the map of the new hallway to allow the participant to find the starting point. (B) After two seconds, the video of the egocentric exploration of the real-world hallway starts. It is presented next to the allocentric map, which remains invariant throughout the trial. (C) In the end of each trial, the person in the video reaches for the door handle of one door. The participant is then asked to indicate the chosen door by tapping on the respective door on the map.

demand on episodic memory, each trial consists of a video of a new hallway section and a respective new map that is presented throughout the trial. Included landmarks were present on both video and map.

Each trial of the paradigm, including instruction and practice, lasts between 18 and 92 seconds ($\bar{M}$ = 47.73 s, $SD$ = 21.01 s), and the 12 main test trials last between 29 and 92 seconds ($\bar{M}$ = 52.92 s, $SD$ = 20.02 s). The trials included zero, one or two 90˚ turns or a full 180˚ turn, and the number of turns increases throughout the test to increase complexity. A turn is defined as a body axis turn of at least 90˚ into a corridor section. Turns towards the chosen door and head motions towards each door during a trial are not considered turns, as they do not alter the perspective of the person in respect to the map. In the first four trials ("no turn" items), the person in the video only goes ahead in sections with increasingly complex layout. The next three trials, "single turn" items 5 to 7, include one turn to the left or right. The trials 8 and 9 ("double turn" items), include two turns to the right (7) or left and right (8). In the three final trials ("full turn" items), the person rotates 180˚ and walks back in the direction they came from.

**Pre-test, instructions and scoring system.** To control for basic perceptual difficulties, the navigation assessment started with a one item pre-test that assessed the participants' ability to translate an egocentric perspective, presented on a two-dimensional screen, to a schematic, allocentric representation of the scene. If the perspective translation pre-test was successfully completed, the navigation test was instructed. Participants were asked to mentally trace the position of the person in the video on the map and indicate their final position. They were told beforehand that the trials increase in complexity and about possible behaviors of the person in the video. During an instruction trial, the main points of the instructions were repeated at an appropriate time of the video. Afterwards, two practice trials were presented and any questions about the task were clarified. Feedback was only given during instruction and practice trials. Exceptions could be made, if giving feedback might prevent a participant from discontinuing the task. It was not possible to repeat a trial and participants were informed that they could take breaks in between trials but not during a trial. When unsure, participants were encouraged to guess the correct door. There was no time limit for responses; however, answers were usually given immediately after the end of a video. A trial was only started following an explicit signal from the participant that they were ready.

Points for the test score were awarded as follows: selection of the correct door was awarded two points; selection of a door directly next to the correct door, or in the same position in a parallel hallway on the same side, was awarded one point because the participant managed the rotation of perspective but the updating was imprecise. In accordance with this, a door directly opposite in the same hallway, or in the same position in a parallel hallway on the opposite side, was also awarded one point because of correct updating but erroneous rotation. All other answers were awarded zero points. For exemplary awarding of points see S1 Fig. Overall performance was rated by the sum of points scored in the 12 main test trials. Thus, test scores could range from 0 to 24 points.

## Data analysis

Data pre-processing, quality check and statistical analyses were performed using the software environment *R* [Version 3.6.1; [49]]. The manuscript was prepared using the *R* package *papaja* [Version 0.1.0.9997; [50]]. For all tests, the level of significance was set at $p < .05$. A visual inspection of the scatterplots of the correlation between the navigation test score and the collected variables did not indicate any non-linear relationships for ordinal or metric variables. The feasibility of the navigation paradigm in the sample was assessed by looking for outliers,

dropouts and participants who had to be excluded because of other reasons than the a priori defined exclusion criteria. The presence of univariate outliers was investigated using the median absolute deviation and transforming the scores into modified *Z*-scores according to Iglewicz and Hoaglin [51]. To identify relevant properties for further analyses, we examined the descriptive statistics and the distribution of the test scores in the novel navigation assessment, as well as subscores of specific item types and other cognitive parameters.

Furthermore, we investigated the statistical relationships of the test performance in the novel navigation assessment with demographic and cognitive variables. We did not include the analyses looking at gender differences, since men were underrepresented in the sample (n = 9, 26%). Due to the small group sizes, secondary education and upper secondary education were combined into one group. For the correlation analyses, the associations with approximately normally distributed ordinal or metric variables were assessed with Pearson's product moment correlation (*r*). Normality was assumed when absolute *z*-scores of both skewness and excess kurtosis were < 1.96, a threshold for small sample sizes [52]. The correlation with ordinal and metric variables without normal distribution in the sample were assessed with Spearman's rho ($r_s$). To control for the effect of metric variables, partial correlations were used. If homogeneity of variance was noted (Levene's test), categorical nominal variables with two categories were assessed with a *t*-test for independent samples. To control for the effect of metric variables in nominal variables, we applied multiple regression analysis with additive effects using $\eta_p^2$ as an effect size measure. A *z*-test was performed when investigating whether one participant differed in their performance.

## Results

### Feasibility

The test was applied without dropouts in all 44 participants for whom data were obtained. Following the standardized verbal instructions, participants completed the task in compliance with the test concept. Four participants reported feeling slightly dizzy during the test and attributed the effect to the camera movement. However, all four participants reported that the mild dizziness ceased during or shortly after test administration. Overall, acquisition was never terminated due to motion sickness.

Statistical analyses in the sample of 34 older adults, after applying the exclusion criteria, further supported the feasibility of the paradigm in healthy seniors. No univariate outliers were identified in the test scores or any of the subscores. All modified *Z*-Scores ($M_i$) of the test score ranged between -1.36 and 1.00, which is below the cut-off for outliers of an absolute value of |$M_i$| > 3.5 [51]. Thus, no participant deviated in their performance compared to the rest of the sample. Furthermore, comparison of the performance of the two participants with impaired color vision to the rest of the sample did not reveal a deviation from the mean performance level (*t*(32) = 0.69, *p* = .494), and the data were therefore retained in further analyses. The participant who was not a native German speaker did not deviate from the mean performance level either (*z* = 0.15, *p* = .881).

### Properties of the navigation assessment

On average, the navigation assessment, including pre-test, instructions and training trials was completed in 18 minutes ($\bar{M}$ = 18.33 min (*SD* = 1.77 min), $\tilde{M}$ = 18 min (MAD = 1.53 min), minimum = 16.02 min, maximum = 24.10 min). Regarding the frequency distribution of the test scores in the novel navigation assessment and the Q-Q plot of the theoretical and sample quantiles, visual inspection indicated normal distribution in the upper half of the possible test

**Table 1. Descriptive statistics of the navigation assessment, its subscores, and other cognitive tests.** Rotation score: items 5–12 with at least one turn; No turn score: items 1–4 without turns; Single turn score: items 5–7 with one 90˚ turn; Double turn score: items 8–9 with two 90˚ turns; Full turn score: items 10–12 with one 180˚ turn.

| Test | $n$ | $\bar{M}$ | $SD$ | $\tilde{M}$ | MAD | Min | Max | Range | $SE$ | $s^2$ | $\gamma_1$ | $z_{\gamma_1}$ | $\gamma_2$ | $z_{\gamma_2}$ |
|---|---|---|---|---|---|---|---|---|---|---|---|---|---|---|
| Navigation assessment | 34 | 18.53 | 3.08 | 18.50 | 3.71 | 11 | 24 | 13 | 0.53 | 9.47 | -0.34 | -0.83 | -0.47 | -0.60 |
| Rotation score | 34 | 11.91 | 2.39 | 12.00 | 2.97 | 7 | 16 | 9 | 0.41 | 5.72 | -0.16 | -0.39 | -1.03 | -1.30 |
| No turn score | 34 | 6.65 | 1.25 | 7.00 | 1.48 | 4 | 8 | 4 | 0.21 | 1.57 | -0.50 | -1.25 | -0.93 | -1.18 |
| Single turn score | 34 | 5.85 | 0.44 | 6.00 | 0.00 | 4 | 6 | 2 | 0.07 | 0.19 | -2.89 | -7.17 | 7.93 | 10.06 |
| Double turn score | 34 | 1.79 | 1.20 | 2.00 | 1.48 | 0 | 4 | 4 | 0.21 | 1.44 | 0.18 | 0.45 | -0.99 | -1.26 |
| Full turn score | 34 | 4.26 | 1.29 | 4.00 | 1.48 | 2 | 6 | 4 | 0.22 | 1.66 | -0.15 | -0.38 | -1.22 | -1.55 |
| Block span forward | 34 | 7.74 | 1.68 | 7.50 | 2.22 | 5 | 12 | 7 | 0.29 | 2.81 | 0.71 | 1.76 | -0.19 | -0.25 |
| Block span backward | 34 | 7.41 | 1.62 | 7.00 | 1.48 | 5 | 11 | 6 | 0.28 | 2.61 | 0.68 | 1.68 | -0.47 | -0.60 |
| Visual learning | 33 | 6.73 | 2.15 | 7.00 | 1.48 | 1 | 10 | 9 | 0.38 | 4.64 | -0.58 | -1.45 | -0.32 | -0.41 |
| Visual recognition | 33 | 7.76 | 2.09 | 8.00 | 1.48 | 2 | 10 | 8 | 0.36 | 4.38 | -0.94 | -2.33 | 0.15 | 0.19 |
| Mental rotation | 34 | 9.35 | 1.45 | 10.00 | 1.48 | 6 | 11 | 5 | 0.25 | 2.11 | -0.38 | -0.93 | -1.07 | -1.36 |

Notes. $\gamma_1$ = skewness, $z_{\gamma_1}$ = Z score skewness, $\gamma_2$ = excess kurtosis, $z_{\gamma_2}$ = Z score excess kurtosis

scores. Only one participant scored the maximum of 24 points. Furthermore, the skewness of -0.34 ($SE_{Skewness}$ = 0.40, 95% CI [-1.13, 0.45]) and kurtosis of -0.47 ($SE_{Kurtosis}$ = 0.79, 95% CI [-2.01, 1.07]) were in the acceptable range [53], satisfying the assumption of normality. Normality was further supported by the absolute $z$-scores of skewness and kurtosis ($z_{Skewness}$ = -0.83, $z_{Kurtosis}$ = -0.60), which were below the threshold of 1.96 for small samples. Table 1 provides an overview of the distribution of the test performance in the novel navigation assessment, the subscores of the specific item types and other cognitive tests.

## Construct validity of the navigation assessment

To evaluate which factors contributed to the performance in the navigation task and in subscores of specific item types, correlations with participant status (age, education, performance in related cognitive tests) were computed (Table 2, S2 Fig). First, we found a large correlation between age and navigation test score ($t(32) = -3.53$, $p = .001$, $r = -.53$), associating poorer performance in the navigation paradigm with increasing age. This was also the case when only items with at least one rotation were considered ($t(32) = -3.36$, $p = .002$, $r = -.51$). Further

**Table 2. Statistical relationship of the test score navigation and item type subscores with demographic and cognitive variables.** Rotation (including items 5–8 with at least one turn); Double (including items 8–9 with two turns); and Full (including items 10–12 with one 180˚ turn). Partial measures are adjusted for participant age.

| | ES | Navigation | Rotation | Double | Full | | Navigation | Rotation | Double | Full |
|---|---|---|---|---|---|---|---|---|---|---|
| Age | $r$ | -.53** | -.51** | -.30 | -.61*** | - | - | - | - | - |
| Education | $d$ | -1.03** | -0.82* | -0.68 | -0.76* | $\eta^2_p$ | 0.28** | 0.19* | 0.11 | 0.19* |
| Block span f | $r$ | .35* | .37* | .44** | .27 | $r_{XY \cdot age}$ | .37* | .40* | .44** | .30 |
| Block span b | $r$ | .36* | .35* | .19 | .41* | $r_{XY \cdot age}$ | .20 | .19 | .08 | .25 |
| Visual learning | $r$ | .31 | .23 | .07 | .33 | $r_{XY \cdot age}$ | .15 | .06 | -.04 | .14 |
| Visual recognition | $r_s$ | .20 | .23 | .10 | .25 | $r_{s_{XY \cdot age}}$ | .09 | .13 | .03 | .15 |
| Mental rotation | $r$ | .30 | .35* | .42* | .21 | $r_{XY \cdot age}$ | .33 | .38* | .43* | .23 |

Notes. Significance levels: *** = $p < .001$, ** = $p < .01$, * = $p < .05$. Effect sizes are: $r$ = Pearson correlation coefficients, $r_s$ = Spearman's rho, $d$ = Cohen's $d$, $\eta^2_p$ is adjusted for participant age, $r_{XY \cdot age}/r_{s_{XY \cdot age}}$ = partial Pearson/ Spearman correlation adjusted for participant age.

scrutiny of the subscores showed that this effect was mostly driven by full turn items ($t(32)$ = −4.31, $p < .001$, $r$ = -.61) since no other subscore correlated significantly with age ($p > .05$).

Furthermore, participants with secondary education ($\bar{M}$ = 16.77) scored lower than those with a high school degree ($\bar{M}$ = 19.62) in the navigation assessment ($t(32)$ = −2.91, $p$ = .007, $d$ = -1.03), as well as in the rotation subscore ($t(32)$ = −2.33, $p$ = .026, $d$ = -0.82), no turn items ($t(32)$ = −2.56, $p$ = .015, $d$ = -0.90), and full turn items ($t(32)$ = −2.15, $p$ = .039, $d$ = -0.76). The multiple regression analyses adjusting for the effect of age showed that participants with high school diploma performed better than participants with secondary education, with $\eta_p^2$ indicating a large effect of education on the navigation score ($t(31)$ = 3.24, $p$ = .003, $\eta_p^2$ = 0.28) and a medium-to-large effect on the rotation score ($t(31)$ = 2.50, $p$ = .018, $\eta_p^2$ = 0.19), no turn items ($t(31)$ = 2.53, $p$ = .017, $\eta_p^2$ = 0.18) and full turn items ($t(31)$ = 2.45, $p$ = .020, $\eta_p^2$ = 0.19).

Regarding the novel assessment's relationship with other cognitive tests, we found medium sized correlations for navigation performance with visuospatial short-term memory ($t(32)$ = 2.08, $p$ = 0.45, $r$ = .35) and working memory ($t(32)$ = 2.16, $p$ = .038, $r$ = .36), assessed by the block span forward and backward, respectively. Higher scores in the block span tasks were associated with higher test scores in the navigation assessment. Neither the markers for visual episodic memory IGD "Visual Learning" and "Visual Recognition," nor mental rotation assessed by the CAV subtest rotation showed a significant correlation with the entire test score ($p > .05$). Considering the subscore rotation, the correlations with visuospatial short-term and working memory were confirmed and a medium-sized correlation with mental rotation was also found ($t(32)$ = 2.11, $p$ = 0.43, $r$ = .35). The direction of the correlations indicates that a higher score in rotation items is associated with a higher score in the mental rotation task. When investigating which item types contributed to these effects, we found that no turn and single turn subscores did not correlate with any of the other cognitive tests ($p > .05$). Double turn scores showed medium-to-large correlations with visuospatial short-term memory ($t(32)$ = 2.77, $p$ = .009, $r$ = .44) and mental rotation ($t(32)$ = 2.65, $p$ = .012, $r$ = .42), while full turn scores showed a medium-to-large correlation with visuospatial working memory ($t(32)$ = 2.56, $p$ = 0.15, $r$ = .41).

Controlling for participant age, partial correlations showed that the correlations of short-term memory with navigation performance ($t(31)$ = 2.23, $p$ = .033, $r_{XY·age}$ = .37), the subscore rotation ($t(31)$ = 2.42, $p$ = .021, $r_{XY·age}$ = .40) and double turn items ($t(31)$ = 2.75, $p$ = .010, $r_{XY·age}$ = .44), as well as the correlation between mental rotation and the subset of rotation items ($t(31)$ = 2.31, $p$ = .028, $r_{XY·age}$ = .38) and double turn items ($t(31)$ = 2.67, $p$ = .012, $r_{XY·age}$ = .43) remained significant. Notably, the relationships with working memory, which were mostly driven by the age sensitive full turn items, were not significant when correcting for the effect of age ($p > .05$).

## Discussion

This work introduced a novel paradigm to assess spatial navigation whose primary strengths include its clear conceptual design with a focus on visuospatial rather than episodic memory abilities, its use of videos of real-life hallways, its standardization, and its quick and intuitive administration in older participants. We provided proof of concept data in healthy, older adults to support its feasibility and construct validity. The paradigm was sensitive to age and education and the successful implementation of the concept was demonstrated through correlations of the navigation test score with short-term and working memory, and partly with mental rotation. Furthermore, we found that age was the driving factor in the association of the navigation assessment with working memory, but did not affect the correlations with short-term memory and mental rotation.

The large negative correlation of age and test performance is in line with previous findings of a decline in performance with older age in navigation abilities in general [7, 8, 54] and more specifically path knowledge [9]. Nevertheless, a strong effect of age implies the need for a large normative sample to evaluate individual performance relative to age. This also applies to the robust effect of education on performance in the novel navigation assessment, even when controlling for participant age. This finding is consistent with the protective effect of education, via increased cognitive reserve, on cognitive decline [55–57] and specifically on navigation ability [58]. Considering this effect of education, future studies using this novel paradigm should examine which cognitive tests show distinct relationships with the navigation assessment beyond the effect of general cognitive ability, which has been shown to correlate with spatial skills [59].

The positive correlations of the navigation assessment scores with visuospatial short-term and working memory, and partly with mental rotation provide evidence for the construct validity of the task. Results from this proof of concept study indicated that visuospatial abilities are more likely to be required than visual episodic memory skills. A closer examination of the role of different item types showed that significant correlations with mental rotation and short-term memory were driven primarily by double turn items and that this finding was robust to correction for the effect of participant age. The correlation of the navigation scores and specifically full turn items with working memory did not remain significant after age correction, which indicates a mechanism how age related changes in working memory performance impact spatial navigation ability in this task. This is in line with findings that specific types of spatial navigation, such as path knowledge, are more sensitive age related changes than others [9] and the mediator effect of executive functions in age related decline for wayfinding tasks [60]. The age sensitivity of working memory has also been related to age related decline in executive functions, specifically updating and inhibition [61], which might explain why the association with short-term memory was not affected by participant age. Taken together, the pattern of relationships of the novel paradigm with demographic variables and other cognitive tests are in line with findings from other spatial navigation tasks and the conceptualization of the passive navigation task.

In this study, no additional requirements for application in healthy older adults without cognitive impairments or subjective complaints emerged. However, the exclusion of the participant with strabismus gives cause for caution when using this paradigm in patients with deficiencies in stereopsis. Beyond spatial vision, it is unlikely that participants with impaired color vision or non-native language levels will be at a disadvantage in the test. This can be inferred from anecdotal evidence in the sample and from a conceptual viewpoint, since the colors of the doors are not indicated on the maps and are therefore unlikely to be a driving factor in task performance. Moreover, since the passive navigation paradigm itself is not language-based, it is likely to be applicable in participants of different nationalities, provided instructions are adequately translated. The normal distribution of the performance in the upper half of the navigation test score range makes it specifically useful in experimental contexts. Although the different item lengths have to be taken into account for timing sensitive experiments, the consistent visual input between participants makes the navigation assessment particularly interesting for studying navigation using magnetic resonance imaging, especially since no head motion is required.

For potential application in patients, the absence of a HID allows unconfounded assessment of spatial navigation in patients with motor impairments, such as tremors, while the short trials and adaptive breaks facilitate application in patients with attention deficits or fatigue. Moreover, the focus on visuospatial rather than episodic memory components of navigation makes it useful to assess spatial navigation in patients with memory impairments, potentially

providing important information for differential diagnostics. However, although floor effects are improbable in patients, the lack of a left-skewed distribution in a healthy population makes it unlikely that the navigation test will be an appropriate screening tool. Taking these aspects into account, the novel navigation assessment is most promising for research settings with older adults and patients. To this effect, the performance level of the navigation paradigm needs to be investigated in different clinical populations with cognitive deficits and subjective cognitive impairment before its sensitivity and diagnostic accuracy for specific disorders can be determined.

Limitations of the novel navigation assessment include the intentional exclusion of route planning and encoding, retrieval or recognition of routes or landmarks, since this limits the paradigm's external validity regarding navigation settings where decisions on an appropriate route or direction have to be made or when navigating previously encountered environments using offline information such as memorized maps. However, a navigation assessment which does not involve an episodic memory task, but still has a high ecological validity and a standardized strategic approach, could help to clarify the role of regions like the hippocampus in navigation, which has been suggested to be involved in navigation via relational memory organization [62]. Especially with regard to unfamiliar environments, where navigation is often performed with all information on the route at hand, including episodic memory components could even be considered less ecologically relevant. Furthermore, the use videos of real-life hallways as opposed to virtual environments contributes to the naturalistic nature of the paradigm, yet also introduces potential distractors such as changes in lighting and limits the flexibility of trial design. Regarding this proof of concept study, the findings need to be replicated in larger and more representative samples, especially in regard to gender, and investigate the role of perspective taking [e.g. PTSOT, 39], executive functions, and selective attention and look at associations with other established active spatial navigation paradigms [e.g. Sea Hero Quest, 7, Cognitive Map Test, 10].

## Conclusions

This work demonstrated that the application of the novel paradigm is promising to study the effect of aging on navigation beyond episodic memory in different clinical settings and foremost experimental studies with older participants. This is facilitated by the navigation test's ecological relevance and absence of an active navigation mechanism, making the assessment quick and intuitive to instruct and administer. Furthermore, it does not require technical resources beyond standard equipment of any clinical or research settings and the navigation assessment's brevity allows an integration into a comprehensive neuropsychological assessment of multiple cognitive functions. The paradigm is age and education sensitive, which has to be taken into account for future studies on the navigations assessment's psychometric properties. While we observed relationships with visuospatial short-term and working memory and partly with mental rotation, no associations with measurements for visual learning or visual recognition were noted, indicating a successful conceptualization and implementation of an easy-to-apply navigation assessment that incorporates core cognitive components of spatial navigation without using an episodic memory task.

## Supporting information

**S1 File. Details on the applied measures.**
(PDF)

**S2 File. Technical details and presentation.**
(PDF)

**S1 Fig.** Exemplary scoring for trials 7 (A) and 9 (B). Green: correct answers were awarded two points. Yellow: doors opposite, next to, or parallel to the correct answer were awarded one point. All other doors were awarded zero points.
(TIF)

**S2 Fig. Scatter plots of the correlations between the navigation assessment scores, age, and cognitive tests.** Scatter plots include regression lines and illustrate the significant correlations of (A) the navigation score with age, block span forward and backward and (B) the rotation score with mental rotation, block span forward and backward. Scatter plots are color coded for participant age.
(TIF)

## Author Contributions

**Conceptualization:** Sophia Rekers.

**Data curation:** Sophia Rekers.

**Formal analysis:** Sophia Rekers.

**Resources:** Michael Niedeggen.

**Supervision:** Michael Niedeggen.

**Visualization:** Sophia Rekers.

**Writing – original draft:** Sophia Rekers.

**Writing – review & editing:** Sophia Rekers, Michael Niedeggen.

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
