## [Decision Letter · Decision Letter 0]

26 Nov 2021

PONE-D-21-24950

Intuitive assessment of spatial navigation beyond episodic memory: Feasibility and proof of concept in middle-aged and elderly individuals

PLOS ONE

Dear Dr. Rekers,

Thank you for submitting your manuscript to PLOS ONE. After careful consideration, we have decided that your manuscript does not meet our criteria for publication and must therefore be rejected. One of the reviewers has raised concersn that cannot be easily addressed by revision. I am sorry that we cannot be more positive on this occasion, but hope that you appreciate the reasons for this decision.

Yours sincerely,

Xiaoang Wan

Academic Editor

PLOS ONE

Reviewers' comments:

Reviewer's Responses to Questions

**Comments to the Author**

1. Is the manuscript technically sound, and do the data support the conclusions?

Reviewer #1: Yes

Reviewer #2: Partly

2. Has the statistical analysis been performed appropriately and rigorously? 

Reviewer #1: Yes

Reviewer #2: Yes

3. Have the authors made all data underlying the findings in their manuscript fully available?

Reviewer #1: Yes

Reviewer #2: Yes

4. Is the manuscript presented in an intelligible fashion and written in standard English?

Reviewer #1: Yes

Reviewer #2: Yes

5. Review Comments to the Author

Reviewer #1: The authors introduced a novel paradigm to assess spatial navigation. It is well designed both conceptually and methodologically. The new paradigm primarily focuses on visuospatial rather than episodic memory abilities. The materials used in the paradigm are videos of real-life hallways and some other high ecological fragments. Most importantly, the paradigm could be easily applied to research in the older population. To my knowledge, it is a novel and innovative work in the area. The experiment is well designed to test the feasibility and validity of the paradigm. Thus, I recommend this work to be published.

Reviewer #2: The present work developed a new paradigm to assess spatial navigation ability in older adults. And authors verified the feasibility and construct validity of the new paradigm, and found para-digm was sensitive to age and education. Specifically, the age of adults influenced the associa-tion of the navigation assessment with working memory as a driving factor.

1， Authors have conducted a lot of relevant analysis in the present study based on the small sample of 34 participants. The result found in this study is not convincing. Thus, I suggest that more participants are needed to prove the validity of the results.

2， In order to make the distribution of each participant’s data clearer, I suggest authors add the scatter plots of correlation results

6. PLOS authors have the option to publish the peer review history of their article (what does this mean?). If published, this will include your full peer review and any attached files.

Reviewer #1: No

Reviewer #2: No

- - - - -

---

## [Author Response · Author response to Decision Letter 0]

20 Jan 2022

Dear Dr. Rekers,

Thank you for submitting your manuscript to PLOS ONE. After careful consideration, we have decided that your manuscript does not meet our criteria for publication and must therefore be rejected. One of the reviewers has raised concersn that cannot be easily addressed by revision. I am sorry that we cannot be more positive on this occasion, but hope that you appreciate the reasons for this decision.

Yours sincerely,

Xiaoang Wan

Academic Editor

PLOS ONE

Response by the authors: From our appraisal of the review, the concern mentioned by Prof. Wan refers to Reviewer #2’s comment on the “small” sample of 34 older participants. We address this point in detail in the section below, where this concern was raised. 

Reviewers' comments:

Reviewer's Responses to Questions

Comments to the Author

1. Is the manuscript technically sound, and do the data support the conclusions?

Reviewer #1: Yes

Reviewer #2: Partly

Response by the authors: From our appraisal of the review, the sole argument raised by Reviewer #2 with regard to soundness of the manuscript concerned the “small” sample of 34 participants. We address this point in detail in the section below, where this concern was raised. 

2. Has the statistical analysis been performed appropriately and rigorously? 

Reviewer #1: Yes

Reviewer #2: Yes

3. Have the authors made all data underlying the findings in their manuscript fully available?

Reviewer #1: Yes

Reviewer #2: Yes

4. Is the manuscript presented in an intelligible fashion and written in standard English?

Reviewer #1: Yes

Reviewer #2: Yes

5. Review Comments to the Author

Reviewer #1: The authors introduced a novel paradigm to assess spatial navigation. It is well designed both conceptually and methodologically. The new paradigm primarily focuses on visuospatial rather than episodic memory abilities. The materials used in the paradigm are videos of real-life hallways and some other high ecological fragments. Most importantly, the paradigm could be easily applied to research in the older population. To my knowledge, it is a novel and innovative work in the area. The experiment is well designed to test the feasibility and validity of the paradigm. Thus, I recommend this work to be published.

Response by the authors: We thank Reviewer #1 for this positive evaluation of our manuscript and the appreciation of the merit of the new paradigm.

Reviewer #2: The present work developed a new paradigm to assess spatial navigation ability in older adults. And authors verified the feasibility and construct validity of the new paradigm, and found paradigm was sensitive to age and education. Specifically, the age of adults influenced the association of the navigation assessment with working memory as a driving factor.

1. Authors have conducted a lot of relevant analysis in the present study based on the small sample of 34 participants. The result found in this study is not convincing. Thus, I suggest that more participants are needed to prove the validity of the results.

Response by the authors: While we agree that 34 participants would not be sufficient for a study aiming to assess the paradigm’s psychometric properties, this was not the objective of this work. As stated in the abstract and throughout the manuscript, this proof of concept study was specifically designed to introduce the concept of the new method, test the feasibility and provide information for future studies ‘[…] investigating the assessment’s psychometric properties in larger samples […]‘.

A sample size of 34 participants is in line with PLOS ONE’s validation criterion for publishing methods: “This requirement may be met by including a proof-of-principle experiment or analysis; if this is not possible, a discussion of the possible applications and some preliminary analysis may be sufficient.” Furthermore, several recent PLOS ONE publications have presented paradigms in behavioral and experimental studies using similar sample sizes (mean n = 24.7, median n = 24, range = 7-47, exact publications listed in the Point-by-point response document).

Lastly, we would like to contend from a statistical point of view that an a priori power analysis indicated that a sample size of 34 was sufficient for detecting statistically significant effects of key variables affecting performance in the novel paradigm, with an effect size of r = .40 (α = .05, power = .80). In fact, post-hoc analyses confirmed that the detected effects referred to in the Comments to the Author by Reviewer #2 achieved statistical powers of .97 for sensitivity to age, .89 for sensitivity to education, and .71 for correlation with working memory.

2. In order to make the distribution of each participant’s data clearer, I suggest authors add the scatter plots of correlation results

Response by the authors: We thank Reviewer #2 for the thoughtful suggestion of adding scatter plots to illustrate significant correlation results between the navigation performance and other cognitive measures, which we added to the manuscript as Figure 3 in the Supporting information. To illustrate the impact of participant age, these are also color coded according to age and effect size, exact p values, and regression lines with confidence intervals are added for easy appraisal of the size and robustness of the effects.

---

## [Editor Report · Decision Letter 1]

14 Jun 2022

Intuitive assessment of spatial navigation beyond episodic memory: Feasibility and proof of concept in middle-aged and elderly individuals

PONE-D-21-24950R1

Dear Dr. Rekers,

We’re pleased to inform you that your manuscript has been judged scientifically suitable for publication and will be formally accepted for publication once it meets all outstanding technical requirements.

Kind regards,

Amir-Homayoun Javadi, PhD

Academic Editor

PLOS ONE

2. Please ensure that you refer to Figure 3 in your text as, if accepted, production will need this reference to link the reader to the figure.

3. Please upload a copy of Supporting Files S1 and S2 which you refer to in your text on pages 10 and 14, respectively.

4. Please update your submission to use the PLOS LaTeX template. The template and more information on our requirements for LaTeX submissions can be found at http://journals.plos.org/plosone/s/latex.

5. We notice that your manuscript file was uploaded on July 30, 2021. Please can you upload the latest version of your revised manuscript as the main article file, ensuring that does not contain any tracked changes or highlighting. This will be used in the production process if your manuscript is accepted. Please follow this link for more information: http://blogs.PLOS.org/everyone/2011/05/10/how-to-submit-your-revised-manuscript/

6.We note that your manuscript is not formatted using one of PLOS ONE’s accepted file types. Please reattach your manuscript as one of the following file types: .doc, .docx, .rtf, or .tex (accompanied by a .pdf).

If your submission was prepared in LaTex, please submit your manuscript file in PDF format and attach your .tex file as “other.”
---

## [Editor Report · Acceptance letter]

30 Jun 2022

PONE-D-21-24950R1 

Intuitive assessment of spatial navigation beyond episodic memory: Feasibility and proof of concept in middle-aged and elderly individuals 

Dear Dr. Rekers:

I'm pleased to inform you that your manuscript has been deemed suitable for publication in PLOS ONE. Congratulations! Your manuscript is now with our production department. 

Kind regards, 

on behalf of

Dr. Amir-Homayoun Javadi 

Academic Editor

PLOS ONE